# TowerVision: Understanding and Improving Multilinguality in Vision-Language Models

## Abstract

Despite significant advances in vision-language models (VLMs), most existing work follows an English-centric design process, limiting their effectiveness in multilingual settings. In this work, we provide a comprehensive empirical study analyzing the impact of several multilingual design choices, such as training data composition, encoder selection, and text backbones. The result is TowerVision, a family of open multilingual VLMs for both image-text and video-text tasks, built upon the multilingual text-only model Tower+. TowerVision achieves competitive performance on multiple multilingual benchmarks and shows particular strength in culturally grounded tasks and multimodal translation. By incorporating visual and cultural context during fine-tuning, our models surpass existing approaches trained on substantially larger datasets, as demonstrated on ALM-Bench and Multi30K (image tasks) and ViMUL-Bench (video tasks). Alongside the models, we release VisionBlocks, a high-quality, curated vision-language dataset. Our findings highlight that multilingual vision-language training data substantially improves cross-lingual generalization—both from high-resource to underrepresented languages and vice versa—and that instruction-tuned LLMs are not always the optimal initialization point. To support further research, we publicly release all models, data, and training recipes.

## 1 Introduction

The success and widespread adoption of large language models (LLMs) has naturally led to a surge of interest in adding multimodal capabilities to these models. In particular, the visual modality has recently received considerable attention, with recent releases of *frontier* vision-language models (VLMs) (Deitke et al., 2024; OpenAI et al., 2024; Comanici et al., 2025; Team et al., 2025; Bai et al., 2025b). However, despite impressive progress, the development of VLMs has been mostly built upon English-centric language models, and trained with English vision-text data, giving little consideration to performance in most other languages. A key challenge in multilingualization of VLMs stems from an asymmetric data landscape—while high-quality *text-only* multilingual corpora are relatively abundant, high-quality multilingual *vision-text* data is scarce. As such, a critical challenge remains: What are the best strategies to effectively extend these models to support multiple languages beyond English?

An effective strategy for VLM multilingualization is to let large-scale text-only multilingual data carry most of the burden. This can be achieved by continuing pretraining of the text backbone on multilingual corpora and by including multilingual content in the text-only portion of the VLM fine-tuning mixture—thereby reducing reliance on scarce multilingual multimodal data. A recent example of this approach is Pangea (Yue et al., 2025), which introduced multilinguality exclusively during the VLM fine-tuning stage using a mixture of data that combined multilingual vision-text pairs generated through synthetic data creation and machine translation of English instructions. While this strategy proved effective, it leaves open key questions: At which stages and on which modules should multilingualization be applied? Which design decisions yield the greatest impact? And how can visual grounding further enhance cross-lingual generalization?

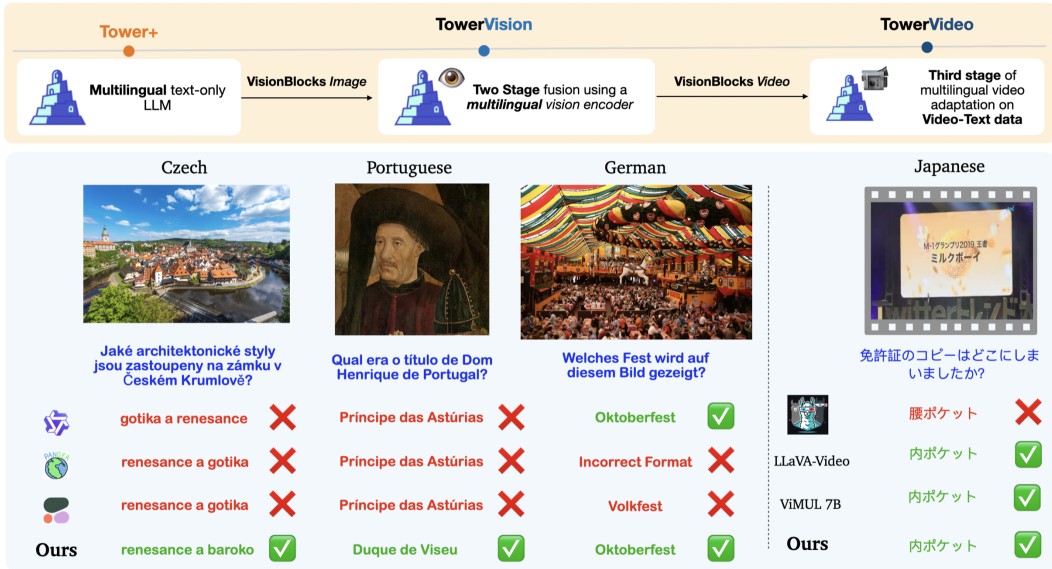

Figure 1: We present TowerVision and TowerVideo, open VLMs with enhanced cultural understanding and multilingual capabilities over leading open multimodal systems on image and video.

In this work, we introduce TowerVision,[1] a suite of open-source multilingual VLMs built on top of Tower+ models (Rei et al., 2025) for 20 languages and dialects.[2] To train TowerVision, we systematically address the challenges outlined above through comprehensive ablation studies, component-level analysis, and cross-lingual evaluation of a multilingualization recipe. Specifically, we investigate how to enhance the multilingual capabilities of VLMs from two axes: first, by exploring the impact of the underlying components (including the alignment projector, vision encoder and text-only LLM); and second, by creating better, more multilingual vision-text datasets and exploring the impact of using this data across different VLM training stages. Overall, compared to strong VLMs of similar size, TowerVision exhibits competitive or superior performance on various multilingual and multimodal benchmarks, as well as cross-lingual transfer capabilities.

In addition to image-based VLMs, we also train a separate multilingual video model, TowerVideo, built on top of TowerVision, thereby extending our analysis to the video modality. TowerVideo achieves competitive performance on ViMUL-Bench (Shafique et al., 2025), a culturally-diverse multilingual video benchmark. Taken together, these contributions provide a comprehensive and systematic study of how to best integrate multilinguality into VLMs across modalities, architectural components, and training stages. Complementing the TowerVision family, we also release VisionBlocks, a curated dataset that consolidates and filters existing vision/video-language resources, further enriched with quality-controlled translations of English textual descriptions into 20 languages and dialects.

## 2 TowerVision

Our approach follows a multi-stage process encompassing three key components, illustrated in Figure 1: (i) a multilingual text-only backbone model, Tower+ Rei et al. (2025); (ii) a Vision Transformer encoder (ViT; Dosovitskiy et al. 2021) that processes visual inputs and extracts meaningful features; (iii) a connector/adapter module that transforms these visual features to generate representations compatible with the text embedding space. These

---

[1] https://huggingface.co/XXX

[2] English, German, Dutch, Spanish (Latin America), French, Portuguese (Portugal), Portuguese (Brazilian), Ukrainian, Hindi, Chinese (Simplified), Chinese (Traditional), Russian, Czech, Korean, Japanese, Italian, Polish, Romanian, Norwegian (Nynorsk) and Norwegian (Bokmål)

modules can be selectively trained or kept frozen during different stages of development (Li et al., 2025). Although this training recipe and variations thereof are well-established and have produced several high-quality models (e.g., LLaVA (Liu et al., 2023b), Intern-VL (Chen et al., 2024), NVLM (Dai et al., 2024), Qwen2.5-VL (Bai et al., 2025b), Molmo (Deitke et al., 2024)), most of these fall short in capturing multilingual and culturally diverse nuances. We therefore introduce our multilingual adaptation, TOWERVISION—we first describe our carefully curated multilingual vision-text data, VISIONBLOCKS (§2.1), and then describe the overall architecture along with an empirically derived recipe, supported by controlled ablations on data allocation, pretraining stages, and initialization strategies(§2.2). (§2.2).

## 2.1 VISIONBLOCKS: TOWARDS BETTER MULTILINGUAL VISION-TEXT DATA

Creating a large-scale, high-quality, multilingual multimodal dataset for training visual language models to be helpful assistants is non-trivial for a series of intertwined reasons:

- *Human-written* vision-text data featuring user-model interactions (common in text-only alignment) is severely limited. While abundant data exists from large-scale captioning datasets (e.g., LAION-5B; Schuhmann et al. 2022), such sources over prioritize scale over quality which is not ideal for training VLMs with advanced capabilities (Dong et al., 2025; Zhou et al., 2023) like instruction-following, helpfulness, and safety.

- High-quality *multilingual* vision-text data is scarce; furthermore, the lack of open, high-quality multilingual VLMs makes controlled synthetic data challenging or restricted to closed models with limited usage licenses. The most viable alternative, also employed by PANGEA (Yue et al., 2025), involves translating English vision-text interactions into target languages.

- Filtering techniques such as reward model scoring or LLM-as-judge approaches (Gu et al., 2025) are significantly more challenging to implement for vision-text data, where even state-of-the-art VLMs (both open and proprietary) struggle to provide reliable preferences (Li et al., 2024).

With this in mind, we develop and release VISIONBLOCKS (Figure 2), which aggregates and filters data from multiple sources, enhanced with new translated and synthetic data, as described below.

**Collection of existing VLM data**  For English vision-text data, we use the mixture created in PIXMO  (Deitke et al., 2024) with a few minor changes: we exclude the Android-Control, Points, and PointQA datasets, as they do not provide additional multilingual value at this stage; For multilingual vision-text data, we leverage a subset of "Open-Ended" and "Multiple-Choice" questions from CULTURALGROUND (de Dieu Nyandwi et al., 2025) and the "Cultural" split of PANGEAINS (Yue et al., 2025) for our languages of interest. The samples from PANGEAINS are originally found in LAIONMulti (Schuhmann et al., 2022) that undergoes a series of automatic steps (using Gemini 1.5 Pro (Gemini Team et al., 2024)) including curating high-quality English instructions, carefully translating them to multiple languages, and adapting them for culturally-relevant multilingual contexts. CULTURALGROUND uses a data curation pipeline that gathers culturally relevant entities from the Wikidata knowledge base, creates several questions and answers about each entity, rephrases them using an LLM, and filters low-quality samples using a VLM. In our work, we rely exclusively on CULTURALGROUND's filtered subsets to ensure maximum quality.

**Translated and synthetic generated vision-language data**  In addition to the original English and multilingual captions, we translate the highly curated PIXMO-CAP caption data Deitke et al. (2024) to our target languages using a TOWER model (Alves et al., 2024). These translations are scored using COMETKIWI  (Rei et al., 2022) and filtered with a high threshold of 0.85 to ensure maximum quality. To further enhance diversity, we pair the remaining high-quality translations with a variety of language-specific captioning prompt templates (§A.5.1). We also augment the dataset with synthetic captions generated by the Gemini 2.5 API. For each image, we sample multiple system prompts to elicit diverse and detailed descriptions (see §A.5.2). This augmentation is intended to improve coverage

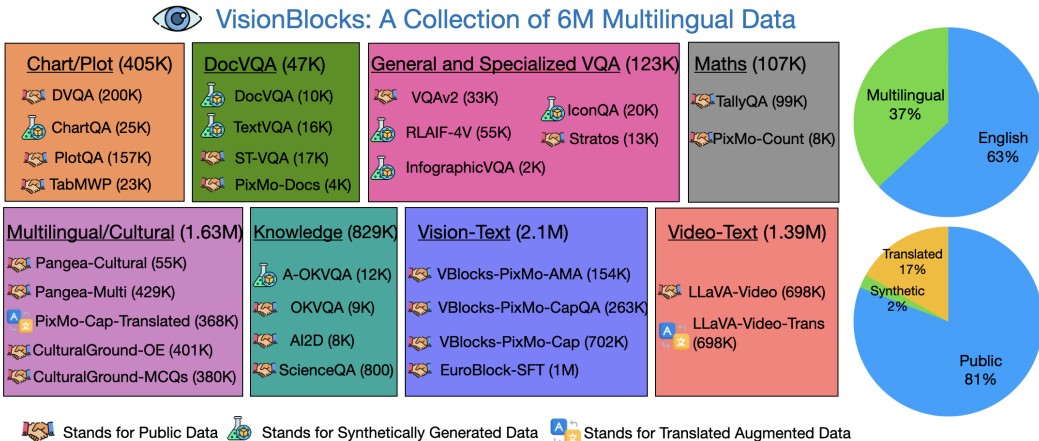

Figure 2: Overview of the VISIONBLOCKS dataset. Synthetic data are generated with Gemini 2.5 API, while translated augmented ones use TOWER (Alves et al., 2024). See Table 8, §A.1 for details.

of fine-grained visual details (e.g., spatial relations, attributes, and contextual cues) that human-authored captions often omit, and provides instruction-like supervision, aligning our model more closely with recent VLM training paradigms that leverage synthetic data to boost generalization and response quality. Similar strategies have been shown to be effective in scaling up instruction-following capabilities of VLMs such as LLaVA (Liu et al., 2023a) and InstructBLIP (Dai et al., 2023). We complete our image-text dataset by incorporating the text-only EUROBLOCKS set, a curated multilingual collection of high-quality synthetic data from the EUROLLM (Martins et al., 2025) synthetic post-training data. EUROBLOCKS provides diverse, instruction-aligned text that enriches our dataset with robust multilingual coverage and fine-grained, high-quality descriptions.

**Translated Multilingual Video Data** As video-text data, we employ the LLaVA-Video-178k dataset (Zhang et al., 2025c), which contains captions alongside open-ended and multiple-choice questions in English. To make the dataset multilingual, we retain a randomly sampled half of the conversations in English, and we translate the remaining half uniformly into all supported languages using TOWER+9B (Rei et al., 2025), thereby ensuring balanced cross-lingual coverage.

## 2.2 TOWERVISION: ARCHITECTURE & TRAINING DETAILS

One way to improve the multilinguality of LLMs (e.g., improving cross-lingual understanding or extending multilingual support for other languages) is to start from a strong pretrained model and continue pretraining on carefully curated data, with subsequent post-training (Xu et al., 2024; 2025; Alves et al., 2024). TOWERVISION follows a similar principle, starting from a strong multilingual Gemma-based backbone TOWER+ 2B/9B (Rei et al., 2025), which achieves strong multilingual general-purpose performance by leveraging a curated high-quality multilingual dataset and a training recipe designed to preserve general capabilities. As shown in §4, starting from this multilingual backbone substantially improves cross-lingual performance compared to starting from Gemma indicating that strong multilingual priors tend to outperform general reasoning models.

For the vision encoder, TOWERVISION is initialized with the recently proposed SigLIP2-so400m/14@384px (Tschannen et al., 2025), a vision transformer operating at $384 \times 384$ resolution that extracts image patch representations and produces multilingually-aligned embeddings of size 729. SigLIP2 is trained on a more diverse data mixture compared to alternatives such as CLIP-ViT (Radford et al., 2021), Perception Encoder (Bolya et al., 2025), or SigLIP1 (Zhai et al., 2023), and thereby yields better multilingual understanding, as we shall see in §4. To align the vision and text modalities, we use a LLaVA-based architecture

(Liu et al., 2023b), where we train a projection layer consisting of a 2-layer MLP, randomly initialized. By combining TOWER+ for text and SigLIP2 for vision, TOWERVISION benefits from complementary multilingual strengths across both modalities. The training process consists of three stages:

1. A *projector pretraining* phase, where we train the model to predict captions given images on the PIXMO-Cap dataset, freezing both the vision encoder and the language model backbone (so only the projector is trained). Each image is encoded once (downscaled to 384×384 if necessary). During this phase, we focus exclusively on diverse, high-quality English captions, which we show to be more effective for aligning visual and textual representations (see §4).

2. A *vision finetuning* phase, where we unfreeze the full model and train it on the full VISIONBLOCKS dataset (§2.1), excluding the video-text data. In this phase, we use *high-dynamic resolution* (Liu et al., 2024a), breaking high-resolution images into a grid of smaller tiles which are then encoded with the vision encoder independently (together with a global thumbnail tile). The projected embeddings are then concatenated. We use a maximum of six tiles, which provides the best trade-off (§A.3). This phase leads to the TOWERVISION model.

3. A *video finetuning* phase, where the video portion of VISIONBLOCKS is used to finetune TOWERVISION on 32-frame video inputs at the encoder's fixed resolution of 384×384. Unlike the previous stage, we omit tiling for efficiency. This phase leads to the TOWERVIDEO model.

The models were trained on a custom fork of the LLaVA-Next (Liu et al., 2024a) codebase.[3]

## 3 EVALUATION & MAIN RESULTS

We evaluate TOWERVISION and TOWERVIDEO on a comprehensive suite of benchmarks spanning multiple modalities and task types (single-image, few-image, and video) across diverse languages, both within and beyond our training set. In this section, we focus on vision-language tasks (i.e., single-image or few image), which including multilingual visual/video question answering, cultural understanding, OCR-related tasks, and visual-language understanding, as well as multilingual video-language tasks. Our assessment relies primarily on closed-form tasks, complemented by large language models serving as judges for video-based evaluations.

### 3.1 TASKS & EVALUATION BENCHMARKS

**Vision-language tasks**   We report results on ALM-Bench (Vayani et al., 2024), a cultural understanding multilingual[4] visual QA benchmark, OCRBench (Liu et al., 2024b) and cc-OCR (Yang et al., 2024) for English and multilingual[5] OCR-centric capabilities respectively, and TextVQA (Singh et al., 2019), assessing scientific understanding. Within cc-OCR, we report results on the multilingual text reading subset, as our primary focus is to evaluate the model's multilingual text recognition capabilities.

**Multimodal translation**   We report results on CoMMuTE (Futeral et al., 2023), a specialized multimodal translation benchmark that uses the visual content to resolve lexical ambiguities present in the source language, and Multi30K (Elliott et al., 2016), a standard benchmark for multimodal machine translation (MT) of image captions.

**Culturally-aware multilingual video tasks**   We use ViMUL-Bench (Shafique et al., 2025), a multilingual video QA benchmark spanning 14 languages: Arabic (ar), Bengali

---

[3]The code will be released upon acceptance.

[4]German, Spanish, French, Italian, Korean, Dutch, Russian, English, Portuguese, Chinese (Simplified and Traditional), Icelandic, Czech, Ukrainian, Hindi, Japanese, Polish, Swedish, Hungarian, Romanian, Danish, Norwegian (Nynorsk), and Finnish.

[5]German, French, Italian, Russian, Spanish, Korean, Portuguese.

Table 1: **Vision-Language Model Performance.** Comparison of English and multilingual VLMs across multiple benchmarks. Reported values correspond to final accuracy (↑). Bold indicates the best score per column. TowerVision results are highlighted.

| | English (↑) | | Multilingual (↑) | | |
|---|---|---|---|---|---|
| | TextVQA | OCRBench | CC-OCR | ALM-Bench (en) | ALM-Bench (multi) |
| Qwen2.5-VL-3B-Instruct | 77.8 | 78.7 | 76.4 | 81.0 | 76.2 |
| Qwen2.5-VL-7B-Instruct | **82.5** | **84.5** | **78.6** | 83.1 | 83.6 |
| Gemma3-4B-it | 65.2 | 74.2 | 69.1 | 79.7 | 80.0 |
| Gemma3-12B-it | 73.2 | 74.7 | 73.8 | 83.5 | 84.5 |
| CulturalPangea7B | 69.8 | 63.5 | 51.7 | 61.3 | 65.2 |
| Llama3-Llava-Next-8B | 64.8 | 54.4 | 40.9 | 76.5 | 73.4 |
| Aya-Vision-8B | 66.9 | 61.0 | 46.3 | 78.2 | 77.3 |
| TowerVision-2B | 68.1 | 58.6 | 46.1 | 77.1 | 81.1 |
| TowerVision-2B-OCR | 69.1 | 63.5 | 55.5 | 76.1 | 77.1 |
| TowerVision-9B | 73.6 | 69.7 | 56.3 | 83.6 | **85.2** |
| TowerVision-9B-OCR | 76.2 | 72.7 | 65.1 | **86.1** | 84.8 |

(bn), Chinese (zh), English (en), French (fr), German (de), Hindi (hi), Japanese (ja), Russian (ru), Sinhala (si), Spanish (es), Swedish (sv), Tamil (ta), and Urdu (ur). The dataset contains both open-ended and multiple-choice questions covering culturally diverse domains such as festivals, customs, food, and heritage. Unlike prior datasets, ViMUL-Bench enables comprehensive evaluation of video-language models across both high- and low-resource languages, promoting inclusive and culturally aware research.

## 3.2 BASELINES

For evaluation, we leverage the lmms-eval framework (Zhang et al., 2025b), which enables a systematic comparison of TOWERVISION against leading open VLMs. We include several multilingual multimodal models, such as *CulturalPangea-7B* (Yue et al., 2025), designed to address gaps in multilingual cultural understanding, and *Aya-Vision-8B* (Singh et al., 2024), optimized for a broad range of vision-language tasks. In addition, we evaluate models from the *Gemma3-Instruct* (*Gemma3-4B-it*, *Gemma3-12B-it*; Team et al. 2025) and the *Qwen2.5-VL-Instruct* families (*Qwen2.5-VL-3B-Instruct*, *Qwen2.5-VL-7B-Instruct*; Qwen et al. 2025), both of which have demonstrated strong performance across a variety of multimodal benchmarks. Finally, we report results for a LLaVA-based model, *Llava-Next-7B* (Liu et al., 2024a), a general-purpose VLM with strong performance across a wide range of tasks. The exact checkpoints for all models are listed in §A.2.

For TOWERVIDEO, we consider several competitive open-source video models of comparable scale, including VideoLLaMA3-7B (Zhang et al., 2025a), LLaVA-Video-7B (Zhang et al., 2025c)—also trained on LLaVA-Video-178k—and ViMUL-7B (Shafique et al., 2025), a multilingual video model.

## 3.3 MAIN RESULTS

Tables 1–2 report the performance of TOWERVISION on vision-language benchmarks as well as multimodal translation benchmarks, while Table 3 reports the results on the multilingual video-language benchmark. We summarize the main findings below.

**TowerVision models are strong in cultural-aware tasks.** Within our suite of vision-language benchmarks, we achieve state-of-the-art results on ALM-Bench (Table 1, a culturally diverse benchmark, in both the English and multilingual split. Qwen2.5VL 7B and Gemma3 12B are the closest competitors, while other baselines lag behind. In the multilingual split, we evaluate on a diverse set of 23 languages covering several language families and scripts. TOWERVISION is able to exhibit enhanced cultural multimodal understanding, suggesting that it is still performant in less seen and unseen languages within its training data. We further assess the cross-lingual generalization capabilities of TOWERVISION in §4.

Table 2: **Multimodal Translation Benchmarks.** We report XCOMET (Guerreiro et al., 2024) for Multi30K and contrastive pairwise accuracy for CoMMuTE. Bold is best.

| | Multi30K (↑) | | | CoMMuTE (↑) | | | |
|---|---|---|---|---|---|---|---|
| | en→cs | en→de | en→fr | en→de | en→fr | en→ru | en→zh |
| Qwen2.5-VL-3B-Instruct | 83.3 | 96.7 | 92.6 | 71.6 | 74.4 | 77.5 | 81.5 |
| Qwen2.5-VL-7B-Instruct | 83.9 | 97.1 | 93.2 | 74.7 | 76.9 | 77.2 | **82.4** |
| Gemma3-4B-it | 33.4 | 44.0 | 33.2 | **76.7** | 78.2 | **79.0** | 74.4 |
| CulturalPangea7B | 80.0 | 95.8 | 92.1 | 68.3 | 77.3 | 75.3 | 79.3 |
| Llama3-Llava-Next-8B | 79.1 | 93.3 | 88.1 | 72.0 | 74.4 | 74.4 | 73.5 |
| Aya-Vision-8B | 94.4 | 97.9 | 95.3 | 69.3 | 76.9 | 74.4 | 76.2 |
| TOWERVISION-2B | 90.3 | 97.5 | 94.7 | 70.0 | 74.3 | 73.2 | 76.6 |
| TOWERVISION-2B-OCR | 90.1 | 97.5 | 94.7 | 70.0 | 77.3 | 74.2 | 76.9 |
| TOWERVISION-9B | **95.1** | 98.1 | 95.6 | 72.0 | **78.8** | 75.6 | 77.4 |
| TOWERVISION-9B-OCR | 94.5 | **98.1** | **95.6** | **72.2** | 78.3 | 75.6 | 77.3 |

Table 3: **Multilingual video performance per language.** Accuracy (%) on ViMUL-Bench across 14 languages averaged across multiple-choice and open-ended questions. Underlined values mark the best score within TOWERVISION/TOWERVIDEO variants; **bold** indicates the best overall. Unsupported languages are marked with *.

| Model | ar | bn* | zh | en | fr | de | hi | ja | ru | si* | es | sv | ta* | ur* |
|---|---|---|---|---|---|---|---|---|---|---|---|---|---|---|
| ViMUL-7B | 41.5 | 35.4 | 37.0 | 48.6 | 48.3 | 43.9 | **39.2** | 37.8 | 45.7 | 21.2 | 44.3 | 41.4 | 23.3 | **36.8** |
| LLaVA-Video-7B | 38.8 | 30.4 | 43.2 | **53.3** | **49.2** | 45.4 | 34.2 | 33.4 | 38.2 | 18.1 | 45.7 | 39.8 | 21.9 | 33.8 |
| VideoLLaMA3-7B | **45.6** | **36.6** | **48.0** | 52.9 | 47.1 | 43.8 | 37.5 | 39.4 | 44.8 | **25.1** | 45.4 | 38.5 | 22.8 | 32.1 |
| TOWERVISION-2B | 18.9 | 19.5 | 21.7 | 34.2 | 28.9 | 28.3 | 25.1 | 22.2 | 24.8 | 16.3 | 30.4 | 27.1 | 16.1 | 19.9 |
| TOWERVIDEO-2B | 23.0 | 18.9 | 35.9 | 45.2 | 39.6 | 39.7 | 37.2 | 34.1 | 38.0 | 17.1 | 37.4 | 38.0 | 17.7 | 18.7 |
| TOWERVISION-9B | 34.2 | 25.4 | 35.3 | 46.7 | 41.1 | 40.8 | 34.2 | 28.1 | 40.3 | 19.8 | 40.5 | 39.6 | 21.6 | 26.4 |
| TOWERVIDEO-9B | 38.6 | 22.1 | 44.8 | 51.9 | 49.1 | **47.1** | 32.2 | **42.3** | 40.9 | 20.8 | **46.0** | **44.8** | **24.1** | 19.5 |

**TowerVision is less competitive on OCR-related tasks.** We hypothesize this is likely due to the limited amount of OCR-focused data in VISIONBLOCKS compared against other models. Since we primarily pretrained TOWERVISION on large-scale image-caption datasets emphasizing natural images and language alignment, it struggles with scanned text or OCR-heavy scenarios. Despite these limitations, TOWERVISION does obtain superior performance compared against Aya Vision 8B and LLaVA Next 8B, the former of which has seen significant amounts of OCR-specific data (Singh et al., 2024).

**TowerVision-2B is competitive multilingually with larger models.** In multimodal translation benchmarks, TOWERVISION consistently demonstrates strong performance on Multi30K and is competitive on CoMMuTE (Table 2). Our 9B variant achieves state-of-the-art results on Multi30k across all language pairs, and we observe that even our smaller 2B variant is a competitive model against the larger baselines on translation-specific, as well as vision-language benchmarks. For instance, on Multi30K, TOWERVISION-2B obtains superior scores to Qwen2.5VL 7B and CulturalPangea 7B. Similarly, on the multilingual split of ALM-Bench, TOWERVISION 2B is competitive with Qwen2.5VL 7B and outperforms Aya Vision 8B. These results further highlight the efficacy of TOWERVISION's multilinguality and design choices. We also note that scaling from 2B to 9B parameters consistently improves performance across all benchmarks, suggesting that our training recipe scales well.

**Multilingual fine-tuning improves cross-lingual performance in TowerVideo.** In Table 3, we report averages across multiple-choice accuracy and open-ended responses, which are automatically judged using GPT-4o (OpenAI et al., 2024), with the same evaluation prompt as Shafique et al. (2025). We compare our TOWERVIDEO models, including the 9B variant, to strong open-source baselines. Our multilingual models are competitive across several languages despite using smaller datasets and fewer frames (for instance, VideoLLaMA3

Table 4: **Impact of backbone and instruction tuning across different benchmarks.**

| Backbone Model | English (↑) | | Multilingual (↑) | | |
|---|---|---|---|---|---|
| | TextVQA | OCRBench | CC-OCR | ALM-Bench (en) | ALM-Bench (multi) |
| Gemma2-pt-2B | 69.2 | 61.2 | 45.3 | 74.3 | 76.7 |
| Tower+pt-2B | **70.3** | 62.1 | **46.3** | 73.0 | 78.2 |
| Gemma2-it-2B | 70.0 | **63.0** | 45.9 | 75.0 | 75.1 |
| Tower+it-2B | 68.1 | 58.6 | 46.1 | **77.1** | **81.1** |
| Gemma2-pt-9B | 72.4 | 66.6 | 49.6 | 79.9 | 79.6 |
| Tower+pt-9B | 73.2 | 64.5 | 54.5 | 81.3 | 84.4 |
| Gemma2-it-9B | **74.4** | 67.2 | 49.5 | 79.6 | 81.5 |
| Tower+it-9B | 73.6 | **69.7** | **56.3** | **83.6** | **85.2** |

uses 180 frames). Specifically, ViMUL was trained with separate copies of the dataset for each language, whereas our approach uses a single copy with half in English and the other half uniformly translated into the supported languages. Overall, these results highlight the effectiveness of video-based multilingual fine-tuning in improving cross-lingual reasoning.

Overall, our results demonstrate the effectiveness of our design choices in endowing our model with strong multilingual capabilities due to a combination of increased multilingual culturally-sensitive training data, a more multilingual text backbone (Tower+), and a multilingual vision encoder. We detail these choices in §4 with a carefully conducted set of ablation experiments.

## 4    Where and How Does Multilinguality Matter?

Following the main results of TowerVision, we delve deeper into its design choices.

**Multilingual backbones improve cross-modal performance.** The choice of backbone in TowerVision can substantially influence performance across multilingual and multi-modal tasks. We focus on two complementary aspects. First, we examine the significance of multilingual capacity by comparing the Tower+ backbone, which is highly multilingual and designed for general-purpose multilingual text tasks, against Gemma2, the model on which Tower+ was built. Second, we investigate the impact of instruction tuning before modality fusion, which is widely applied in modern VLMs from the start (Liu et al., 2023b; Bai et al., 2025a), but whose effect on the final model remains unclear. To study these effects, we train TowerVision at 2B and 9B scales using three backbones: Gemma2-pt (pre-trained, not instruction-tuned), Tower+pt (pretrained Tower+, not instruction-tuned), and Tower+it (instruction-tuned Tower+), following the recipe in §2. As shown in Table 4, using Tower+ consistently outperforms Gemma2, confirming the importance of a multilingual backbone for robust cross-modal understanding. At smaller scales, non-instructed models (Gemma2-pt, Tower+pt) retain stronger raw visual extraction, while instruction-tuned variants excel in cultural knowledge and reasoning. By the 9B scale, this gap narrows, with instruction-tuned models integrating both skills and achieving state-of-the-art performance. These findings underscore the complementary roles of multilingual pretraining and instruction tuning, and the need for careful backbone selection in VLMs.

**Multilingual-aware vision encoders improve performance in low-data regimes.** Effectively leveraging multilingual data is crucial for VLMs, yet it is unclear whether the vision encoder's own multilingual capacity plays an important role. We compare SigLIP2, trained on diverse multilingual data, with SigLIP1, an earlier English-centric version, to test whether multilingual-aware encoders are essential or if sufficient fine-tuning can compensate. We train TowerVision with both encoders on English-only and multilingual data at 2B and 9B scales (results in Table 5).

Without additional multilingual data, SigLIP2 models consistently outperform SigLIP1, showing clear benefits in low data regimes, where training data is scarce. With multilingual fine-tuning, however, the gap narrows, showing that finetuning with sufficient multilingual data can compensate for a weaker encoder. At 9B scale, both converge to strong perfor-

Table 5: Multilingual impact of different vision encoders measure on ALM-Bench.

| TowerVision | 2B | | 9B | |
|---|---|---|---|---|
| Variant | En | Multi | En | Multi |
| SigLIP1-En | 67.4 | 60.2 | 78.3 | 81.2 |
| SigLIP2-En | 69.3 | 67.1 | 77.2 | 81.1 |
| SigLIP1-(En+Multi) | 76.6 | 80.7 | 83.6 | 84.4 |
| SigLIP2-(En+Multi) | **77.1** | **81.1** | **83.6** | **85.2** |

mance. In short, multilingual-aware encoders provide an advantage when data is scarce, but extensive multilingual training can close the gap.

**High-quality English captions are enough to ensure strong alignment.** To assess whether multilingual supervision is necessary during alignment pretraining, we train two versions of TowerVision on both scales, 2B and 9B.

The first version uses only English-only captions from PixMo-Cap, comprising 702, 205 text-image pairs. The second version uses the same English captions combined with a high-quality translated subset from PIXMO-CAP, where data was uniformly translated into the supported languages as described in §2.1, comprising 367,779 samples. We evaluate the models in ALM-Bench to measure TowerVision performance both in English and across multiple non-English languages, providing insights into how well cross-lingual generalization is preserved or improved. As shown in Table 6, adding high-quality multilingual captions during the projector alignment stage has little to no positive effect and, in some cases, slightly decreases performance on the multilingual subset. This suggests that the most effective strategy is to focus on diverse and high-quality captions, ensuring strong alignment between visual and textual modalities, rather than prioritizing extensive multilingual coverage at this stage.

Table 6: Effect of using multilingual versus English-only captions during projector pretraining on ALM-Bench. Results indicate low to no gains from adding multilingual data at this stage.

| TowerVision | 2B | | 9B | |
|---|---|---|---|---|
| Projector | En | Multi | En | Multi |
| En | 77.1 | **81.1** | **83.6** | **85.2** |
| En+Multi | **77.9** | 79.3 | 83.0 | 84.1 |

**Expanding languages improves cross-lingual generalization in VLMs.** We study how language coverage in training data impacts performance on both included and excluded languages. Specifically, we compare training on 10 high-resource "core languages" versus the full set of languages, while controlling for dataset size. Our questions are: (i) whether adding balanced multimodal data for more languages improves performance on core languages (Conneau et al., 2020; Hu et al., 2020), and (ii) whether unsupported languages benefit in zero-shot fashion if related languages are present (Ni et al., 2021). We train TowerVision at 2B and 9B scales using the recipe in §2, first on 10 "core" languages (English, German, Dutch, Portuguese, Russian, Simplified and Traditional Chinese, Spanish, French, Italian), then on all available languages. Results in Figure 3 (more details in §A.4) show that broader language coverage consistently improves performance, with larger gains at the 2B scale. Zero-shot improvements for unsupported languages further support cross-lingual transfer when related languages are included. These findings highlight the value of expanding multilingual data, particularly for smaller models.

**How does multilingual data affect video fine-tuning?** To assess the impact of our multilingual data (see § 2.1) during video fine-tuning, we present results in Table 7 for two baselines: (i) the original TowerVision-2B model and (ii) TowerVideo-2B trained on the full English-only LLaVA-Video-178k dataset. Fine-tuning with video substantially improves the performance of TowerVision models compared to image-text-only variants, highlighting the importance of temporal information for video-language understanding. Incorporating multilingual data further enhances cross-lingual generalization, while English performance remains largely stable, indicating that adding multiple languages does not com-

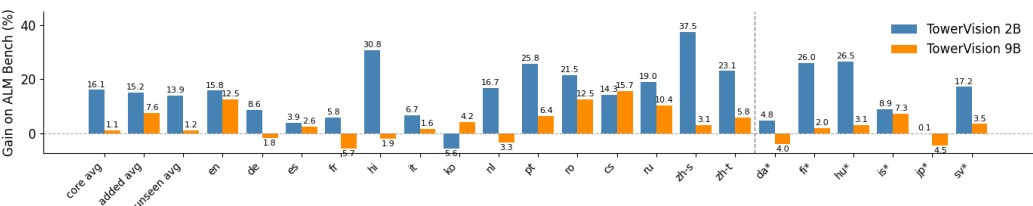

Figure 3: Performance of TowerVision models on 10 vs 20 languages/dialects at 2B and 9B scales. The bars indicate the accuracy gains by training on 20 (all) versus 10 (core) languages.

Table 7: Accuracy (%) on ViMUL-Bench across 14 languages averaged across multiple-choice and open-ended questions. Underlined values mark the best score within TowerVision/TowerVideo variants; **bold** indicates the best overall. Unsupported languages are marked with *.

| Model | ar | bn* | zh | en | fr | de | hi | ja | ru | si* | es | sv | ta* | ur* |
|---|---|---|---|---|---|---|---|---|---|---|---|---|---|---|
| TowerVision-2B | 18.9 | **19.5** | 21.7 | 34.2 | 28.9 | 28.3 | 25.1 | 22.2 | 24.8 | 16.3 | 30.4 | 27.1 | 16.1 | **19.9** |
| TowerVideo-2B (english only) | **25.7** | 17.8 | 26.7 | **45.5** | **42.3** | 34.8 | 27.8 | 27.7 | 34.4 | **17.9** | **37.8** | 34.0 | **18.3** | 19.7 |
| TowerVideo-2B (multilingual) | 23.0 | 18.9 | **35.9** | 45.2 | 39.6 | **39.7** | **37.2** | **34.1** | **38.0** | 17.1 | 37.4 | **38.0** | 17.7 | 18.7 |

promise primary-language capabilities, even though the multilingual models are trained on substantially less English data.

## 5 CONCLUSION

We introduced TowerVision, a suite of multimodal models for image-text and video-text tasks, designed with a strong emphasis on cultural understanding and multilinguality. Our models demonstrate competitive, and in several cases improved, multilingual performance across a range of benchmarks when compared with existing open multimodal systems. Alongside this, we released VisionBlocks, a high-quality vision-language dataset, and provided a detailed training recipe covering data, encoders, and text backbones, complemented by an extensive ablation study on key components of our approach.

We hope that these contributions—spanning models, data, and methodology—help advance research on culturally diverse multilingual multimodal language models, and accelerate progress toward narrowing the performance gap with English-centric settings.

## 6 ETHICS STATEMENT

This work develops and evaluates multilingual vision-language models using publicly available datasets as well as our own synthetic and translated data. We acknowledge potential risks, including biased model outputs and unintended misuse of generated content. While we have taken steps to ensure diversity and maximum data quality, we always encourage careful evaluation and responsible deployment of these models in real-world scenarios. Our research does not involve sensitive personal data or tasks with direct safety-critical impact.

## 7 REPRODUCIBILITY STATEMENT

This work provides detailed descriptions of the data, model architectures, training procedure (including the codebase), and evaluation benchmarks used. All datasets used are either publicly available or created by our team (synthetic and translated), with the respective system prompts shared for maximum transparency. Additionally TowerVision all the collection of models, code for data preprocessing, training, and evaluation will be released

to facilitate replication of our results. We aim to ensure that other researchers can reproduce our findings with minimal effort.

We ensure reproducibility by providing detailed descriptions of the data, model architectures, training procedures, and evaluation benchmarks. Upon acceptance, we will release the VISIOBLOCKS dataset[6], checkpoints of the TOWERVISION collection models[7], and the corresponding codebases for training and evaluation[8], to facilitate replication of our results.

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

Table 8: Overview of dataset composition across categories. Each dataset lists its sample size with the proportion of the total in parentheses, along with its collection type tag (Public Data, Synthetic (Generated), or Translated (Augmented)). Totals are shown for English-only and Multilingual subsets, as well as the overall dataset size.

| Category | Dataset | Samples (%) | Tag |
|---|---|---|---|
| Chart/Plot | DVQA | 199,995 (3.17%) | Public Data |
| | ChartQA | 25,055 (0.40%) | Synthetic (Generated) |
| | PlotQA | 157,070 (2.49%) | Public Data |
| | TabMWP | 22,717 (0.36%) | Public Data |
| General VQA | VQAv2 | 428,708 (6.79%) | Public Data |
| | RLAIF-4V | 59,408 (0.94%) | Synthetic (Generated) |
| Doc VQA | DocVQA | 9,664 (0.15%) | Synthetic (Generated) |
| | TextVQA | 15,690 (0.25%) | Synthetic (Generated) |
| | ST-VQA | 17,242 (0.27%) | Public Data |
| | PixMo-Docs | 3,634 (0.06%) | Public Data |
| Reasoning/Knowledge | A-OKVQA | 11,853 (0.19%) | Synthetic (Generated) |
| | OKVQA | 9,009 (0.14%) | Public Data |
| | AI2D | 7,791 (0.12%) | Public Data |
| | ScienceQA | 758 (0.012%) | Public Data |
| Multilingual/Cultural | Pangea-Cultural | 55,438 (0.88%) | Public Data |
| | Pangea-Multi | 428,838 (6.79%) | Public Data |
| | PixMo-Cap-Translated | 367,779 (5.83%) | Translated (Augmented) |
| | CulturalGround-OE | 401,149 (6.35%) | Public Data |
| | CulturalGround-MCQs | 379,834 (6.02%) | Public Data |
| Specialized VQA | IconQA | 19,543 (0.31%) | Synthetic (Generated) |
| | InfographicVQA | 2,049 (0.03%) | Synthetic (Generated) |
| | Stratos | 12,585 (0.20%) | Public Data |
| Counting/Math | TallyQA | 98,675 (1.56%) | Public Data |
| | PixMo-Count | 8,128 (0.13%) | Public Data |
| Vision/Text | VBlocks-PixMo-AMA | 154,336 (2.44%) | Public Data |
| | VBlocks-PixMo-Cap | 702,205 (11.12%) | Public Data |
| | VBlocks-PixMo-CapQA | 262,862 (4.16%) | Public Data |
| | EuroBlocks-SFT | 1,094,265 (17.34%) | Public Data |
| Video/Text | LLaVA-Video-178k-subset | 697,618 (11.05%) | Public Data |
| | LLaVA-Video-178k-translated | 697,617 (11.05%) | Translated (Augmented) |
| | **Total (English)** | 3,982,630 (63.1%) | |
| | **Total (Multilingual)** | 2,330,656 (36.9%) | |
| | **Overall Total** | 6,313,286 (100%) | |

# A  APPENDIX

## A.1  FULL DESCRIPTION OF VISIONBLOCKS

Table 8 shows the full details and statistics of the VISIONBLOCKS dataset.

## A.2  MODELS CHECKPOINTS

Table 9 lists all model checkpoints used for comparative baselines. We use checkpoints released HuggingFace when possible.

## A.3  VISION ENCODER VARIANTS

Beyond selecting a more multilingual vision encoder, several other factors significantly influence its performance. These include the input image resolution supported by the encoder, the number of patches it uses, which determines the total number of visual tokens for a

| Model | Params | Checkpoint Link |
|---|---|---|
| Qwen2.5-VL-Instruct | 3B | `https://huggingface.co/Qwen/Qwen2.5-VL-3B-Instruct` |
| Qwen2.5-VL-Instruct | 7B | `https://huggingface.co/Qwen/Qwen2.5-VL-7B-Instruct` |
| Gemma2-it | 2B | `https://huggingface.co/google/gemma-2-2b-it` |
| Gemma2-pt | 2B | `https://huggingface.co/google/gemma-2-2b` |
| Gemma2-it | 9B | `https://huggingface.co/google/gemma-2-9b-it` |
| Gemma2-pt | 9B | `https://huggingface.co/google/gemma-2-9b` |
| Gemma3-it | 4B | `https://huggingface.co/google/gemma-3-4b-it` |
| Gemma3-it | 12B | `https://huggingface.co/google/gemma-3-12b-it` |
| CulturalPangea | 7B | `https://huggingface.co/neulab/CulturalPangea-7B` |
| LLaVA-Next | 7B | `llava-hf/llava-v1.6-mistral-7b-hf` |
| Aya-Vision | 8B | `https://huggingface.co/CohereForAI/aya-vision-8b` |
| Pixtral | 12B | `https://huggingface.co/mistralai/Pixtral-12B-2409` |
| Phi-4-Multimodal | 14B | `https://huggingface.co/microsoft/Phi-4-multimodal-instruct` |

Table 9: **Model checkpoints.** Parameters and HuggingFace links for models included in our evaluation suite.

given image resolution (e.g, for an img resolution of $224 \times 224$ using patch size of 14 we obtain 256 visual tokens) and the number of tiles.

Our goal is to empirically identify the optimal configuration for processing visual inputs, focusing on these three factors.

Specifically, we perform experiments using the TowerVision 2B version with variants of SigLIP2 framework:

1. Image resolution: We vary the input image size between $384 \times 384$, $224 \times 224$, and $512 \times 512$ to examine its effect on feature extraction quality.

2. Patch numbers: We test different patch sizes (14 and 16) to assess how granularity impacts the learned representations. Smaller patches capture finer details but increase the number of tokens, affecting the context length the model must handle.

3. Number of tiles: Beyond the default 6 tiles, we also experiment with 4 and 22 tiles. The number of tiles is adjusted to the image resolution: lower-resolution images (e.g, $224 \times 224$) require more tiles to cover the same amount of visual information as a higher-resolution encoder (e.g., $512 \times 512$). For example, an image with resolution $(1024, 1024)$ processed with a $512 \times 512$ encoder requires roughly 4 tiles to cover the full image, whereas a $224 \times 224$ encoder would need at least 25 tiles (including padding) to achieve similar coverage. This creates a trade-off between capturing detailed local information and maintaining manageable context length.

These experiments allow us to systematically compare variations while keeping other components constant, providing insights into which configuration yields the best overall performance. Results are reported in Table 10, highlighting the trade-offs between resolution, patch granularity, and style diversity.

## A.4 Cross-Lingual Generalization

## A.5 System Prompts

### A.5.1 Tower System Prompts used for Translation

The prompts vary in style and specificity to improve diversity and capture nuanced meaning from the original English captions. They are grouped by language and include multiple phrasings for the same instruction to encourage robust translations.

Table 10: **Impact of Vision Encoder Configuration and Instruction Tuning.** Evaluation of TOWER+ models across English and multilingual tasks with varying image resolution, patch size, and number of tiles. Results highlight how these design choices affect overall performance.

| Resolution | Patch Size | Tiles | English | | Multilingual | |
|---|---|---|---|---|---|---|
| | | | TextVQA | OCRBench | CC-OCR | ALM-Bench |
| 224x224 | 14 | 22 | 59.1 | 53.3 | 37.2 | 70.5 |
| 224x224 | 16 | 20 | 68.6 | 57.8 | 44.3 | 75.2 |
| 384x384 | 14 | 6 | **70.3** | **62.1** | **46.1** | **75.6** |
| 512x512 | 16 | 4 | 64.0 | 55.7 | 39.6 | 74.7 |

Table 11: Cross-lingual performance of TOWERVISION models at 2B and 9B scales, evaluated on the ALM-Bench benchmark. *Core Langs* refers to a set of 10 languages: English, German, Dutch, Portuguese, Russian, Simplified and Traditional Chinese, Spanish, French and Italian. *Core+Added Langs* includes all languages supported by TOWERVISION as indicated in footnote 2. *Unseen* languages are those not encountered during training and are marked with an asterisk (*). Bold values indicate the better result within each scale. Positive gains from adding languages are highlighted in light green, negative gains in light red.

Overall, adding more languages tends to improve performance across the board, demonstrating strong cross-lingual transfer capabilities, even for unseen languages.

| Metric / Lang | TowerVision-2B | | | TowerVision-9B | | |
|---|---|---|---|---|---|---|
| | Core Langs | Core + Added Langs | Gain | Core Langs | Core + Added Langs | Gain |
| English (en) | 60.9 | **76.6** | +15.8 | 70.3 | **82.8** | +12.5 |
| Core Avg | 65.3 | **81.3** | +16.1 | 81.5 | **82.6** | +1.1 |
| Added Avg | 60.2 | **75.4** | +15.2 | 76.3 | **84.3** | +7.6 |
| Unseen Avg | 69.2 | **83.0** | +13.9 | 81.2 | **82.5** | +1.2 |
| German (de) | 75.9 | **84.5** | +8.6 | **89.7** | 87.9 | -1.8 |
| Spanish (es) | 56.6 | 60.5 | +3.9 | 73.7 | **76.3** | +2.6 |
| French (fr) | 76.9 | **82.7** | +5.8 | **86.5** | 80.8 | -5.7 |
| Hindi (hi) | 44.2 | **75.0** | +30.8 | 82.7 | 80.8 | -1.9 |
| Italian (it) | 75.0 | **81.7** | +6.7 | 96.7 | **98.3** | +1.6 |
| Korean (ko) | **76.4** | 70.8 | -5.6 | 75.0 | **79.2** | +4.2 |
| Dutch (nl) | 70.0 | **86.7** | +16.7 | **90.0** | 86.7 | -3.3 |
| Portuguese (pt) | 64.5 | **90.3** | +25.8 | 85.5 | **91.9** | +6.4 |
| Romanian (ro) | 58.9 | **80.4** | +21.5 | 75.0 | **87.5** | +12.5 |
| Czech (cs) | 61.4 | **75.7** | +14.3 | 74.3 | **90.0** | +15.7 |
| Russian (ru) | 65.5 | **84.5** | +19.0 | 65.5 | **75.9** | +10.4 |
| Chinese (simp.) (zh-hans) | 50.0 | **87.5** | +37.5 | 68.8 | 71.9 | +3.1 |
| Chinese (trad.) (zh-hant) | 53.8 | **76.9** | +23.1 | 61.5 | 67.3 | +5.8 |
| Danish (da)* | 66.1 | 70.9 | +4.8 | **90.3** | 86.3 | -4.0 |
| Finnish (fi)* | 56.0 | **82.0** | +26.0 | 70.0 | 72.0 | +2.0 |
| Hungarian (hu)* | 68.8 | **95.3** | +26.5 | 79.7 | **82.8** | +3.1 |
| Icelandic (is)* | 67.6 | **76.5** | +8.9 | 76.5 | **83.8** | +7.3 |
| Japanese (jp)* | **78.8** | 78.9 | 0.1 | **84.8** | 80.3 | -4.5 |
| Swedish (sv)* | 77.6 | **94.8** | +17.2 | 86.2 | **89.7** | +3.5 |

```python
# English prompts
EN_PROMPTS = [
    "Describe this image.",
    "What can you see in this picture?",
    "Tell me what's in this image.",
    "Explain what this image shows.",
    "Caption this image.",
    "What's happening in this picture?",
    "Provide a description of this image."
]

# European Portuguese prompts
PT_PROMPTS = [
```

```
        "Descreva esta imagem.",
        "O que consegue ver nesta fotografia?",
        "Diga-me o que está nesta imagem.",
        "Explique o que esta imagem mostra.",
        "Legende esta imagem.",
        "O que se passa nesta fotografia?",
        "Forneça uma descrição desta imagem."
    ]

    # French prompts
    FR_PROMPTS = [
        "Décrivez cette image.",
        "Que pouvez-vous voir sur cette photo?",
        "Dites-moi ce qu'il y a dans cette image.",
        "Expliquez ce que cette image montre.",
        "Légendez cette image.",
        "Que se passe-t-il sur cette photo?",
        "Fournissez une description de cette image."
    ]

    # Dutch prompts
    NL_PROMPTS = [
        "Beschrijf deze afbeelding.",
        "Wat zie je op deze foto?",
        "Vertel me wat er op deze afbeelding staat.",
        "Leg uit wat deze afbeelding laat zien.",
        "Onderschrift deze afbeelding.",
        "Wat gebeurt er op deze foto?",
        "Geef een beschrijving van deze afbeelding."
    ]

    # German prompts
    DE_PROMPTS = [
        "Beschreiben Sie dieses Bild.",
        "Was können Sie auf diesem Foto sehen?",
        "Sagen Sie mir, was auf diesem Bild zu sehen ist.",
        "Erklären Sie, was dieses Bild zeigt.",
        "Beschriften Sie dieses Bild.",
        "Was passiert auf diesem Foto?",
        "Geben Sie eine Beschreibung dieses Bildes."
    ]

    # Spanish prompts
    ES_PROMPTS = [
        "Describe esta imagen.",
        "¿Qué puedes ver en esta foto?",
        "Dime qué hay en esta imagen.",
        "Explica qué muestra esta imagen.",
        "Pon un título a esta imagen.",
        "¿Qué está pasando en esta foto?",
        "Proporciona una descripción de esta imagen."
    ]

    # Italian prompts
    IT_PROMPTS = [
        "Descrivi questa immagine.",
        "Cosa puoi vedere in questa foto?",
        "Dimmi cosa c'è in questa immagine.",
        "Spiega cosa mostra questa immagine.",
        "Dai un titolo a questa immagine.",
        "Cosa sta succedendo in questa foto?",
        "Fornisci una descrizione di questa immagine."
    ]
```

```
# Korean prompts
KO_PROMPTS = [
    "이 이미지를 설명해주세요.",
    "이 사진에서 무엇을 볼 수 있나요?",
    "이 이미지에 무엇이 있는지 알려주세요.",
    "이 이미지가 보여주는 것을 설명해주세요.",
    "이 이미지에 캡션을 달아주세요.",
    "이 사진에서 무슨 일이 일어나고 있나요?",
    "이 이미지에 대한 설명을 제공해주세요."
]

# Chinese prompts
ZH_PROMPTS = [
    "描述这张图片。",
    "你能在这张照片中看到什么？",
    "告诉我这张图片里有什么。",
    "解释这张图片展示了什么。",
    "为这张图片添加说明。",
    "这张照片中发生了什么？",
    "提供这张图片的描述。"
]
```

### A.5.2 GEMINI 2.5 SYSTEM PROMPTS

We generate synthetic captions using the Gemini 2.5 API with a diverse set of system prompts. These prompts are designed to produce varied response formats, including direct answers, caption-plus-answer pairs, and structured final-answer formats.

```
# Direct answer formats
    "Answer the question concisely.",
    "Provide a brief, direct answer to the question.",
    "Keep your response short and to the point.",
    "Give a concise answer based on what you see in the image.",
    "Answer directly based on the visual information.",
    "Respond with a short, clear answer to the question.",
    "Be brief and direct in your response."

# Simple caption + answer formats
    "First provide a caption of what you see, then give your answer.",
    "Write a brief caption describing the image, followed by your answer to the question.",
    "Start with a description of the image, then provide your answer clearly marked as 'Answer:'.",
    "First write 'Caption: <brief image description>' then answer the question.",
    "Begin with 'Caption: [what you see in the image]' followed by your response to the question.",
    "Start by writing 'CAPTION: {description}' before answering the question."

# Final Answer formats
    "End your response with 'Final Answer: <your answer>'.",
    "Conclude with 'Final Answer: <your answer>'.",
    "After looking at the image, provide 'Final Answer: <your answer>'.",
    "Your response should end with 'Final Answer: <your answer>'.",
    "First describe what you see, then provide 'Final Answer: <your answer>'.",
    "Always end your response with 'Final Answer: <your answer>' after analyzing the image.",
    "Provide a concise answer. End with 'Final Answer: <your answer>'."

# Naive formats (simple, direct)
    "Describe the image and answer the question.",
    "Begin by describing the image and then answer the question.",
    "Provide a brief description of the image and then answer the question.",
    "Answer the question in a helpful and informative manner.",
    "Start by describing the image and then answer the question.",
    "You are a helpful assistant. Describe the image and answer the question."
```

```
# Simple formatted caption/answer pairs
    "Caption: <description> → Answer: <response>",
    "Image shows: <description> | My answer: <response>",
    "[CAPTION] <description> [ANSWER] <response>",
    "# Image: <description>\n# Answer: <response>",
    "First 'Image Description: <what you see>' then 'Answer: <your response>'"

# With specific markers
    "<description><answer>",
    "Image: <description> → Answer: <conclusion>",
    "<IMAGE> describe what you see </IMAGE> <ANSWER> provide your response </ANSWER>"
"Begin with '{IMAGE DESCRIPTION}' and end with '{FINAL ANSWER}'."
```

These prompts are used to generate high-quality captions that improve instruction-following and visual description diversity.

