# OpenReview forum: "TowerVision : Understanding and Improving Multilinguality in Vision-Language Models"
_ICLR.cc/2026/Conference — ICLR 2026 Conference Withdrawn Submission_

### Official Review · Reviewer_yWBs · 2025-10-20

**Soundness:** 3
**Presentation:** 3
**Contribution:** 2
**Rating:** 4
**Confidence:** 3

**Summary:**

This paper presents TowerVision, a suite of multimodal multilingual large language models (LLMs) for both image-text and video-text tasks, built upon the Tower+ models and SigLIP2. TowerVision leverages visual and cultural context during fine-tuning through a three-stage training process consisting of: (1) projector pretraining, (2) vision fine-tuning, and (3) video fine-tuning. The final TowerVision models outperform existing systems trained on much larger datasets across benchmarks such as ALM-Bench, Multi30K, and ViMUL-Bench.

Key contributions of the paper include the release of VISIONBLOCKS, a curated multilingual vision-language dataset; the open-sourcing of the models; and empirical insights into model backbones and training data composition for building more effective multilingual multimodal LLMs.

**Strengths:**

S1: The paper provides a suite of open-sourced multimodal multilingual LLMs for both image-text tasks, as well as video text tasks. The paper also provides a curated training dataset including both human and synthetically generated resources of 6M instances for enabling future research in this area.

S2: The evaluation of Towervision is extensive. They span across multiple modalities, languages (20+), and task types (VQA, OCR, translation, cultural reasoning). The resulting models are competitive against strong baselines (Qwen2.5-VL, Gemma3, Aya-Vision, CulturalPangea) and demonstrate robust performance.

S3: The analysis of where multilingual training matters is insightful, where multilingual backbones matter more than multilingual captions during alignment.

**Weaknesses:**

W1: While the paper provides an analysis of when multilingual training can improve the final LLM’s performance, a deeper examination of the data mixture would offer additional insights. For instance, the authors could analyze the benefits of training on different types of datasets (e.g., mathematical versus chart-based datasets).


W2: Although the paper addresses an important problem and contributes new models to multilingual multimodal research, it offers limited architectural and methodological novelty.

W3: The clarity of writing could be improved, as some important details are missing. For example:
 - Table 4: The Gemma2-2B/9B-IT models are not included in the comparison. How do the results compare to fine-tuning the IT model?
 - Table 5: Details about the dataset used for evaluation are missing. What data was used for these evaluations?
 - Line 398: The term “low-data regimes” should be clarified. What exactly does it refer to in this context?
 - Table 7: How does multilingual data affect video fine-tuning for the 9B model?

**Questions:**

- See W3
- Do you have any insights regarding to the data mixture? What type of training data is effective in improving the ViMUL-Bench?

---

> ### Author Response · Authors · 2025-11-21
> **Answer to reviewer yWBs (part1)**
>
> Thank you for your insightful comments.
>
> > While the paper provides an analysis of when multilingual training can improve the final LLM’s performance, a deeper examination of the data mixture would offer additional insights. For instance, the authors could analyze the benefits of training on different types of datasets (e.g., mathematical versus chart-based datasets).
>
> This is a great suggestion. We did not perform an extensive ablation over all possible data mixtures, as this would be computationally intensive and beyond our available resources. To manage these demands, we focused on the core premise of our work: understanding the impact of different ratios of multilingual data in multilingual multimodal tasks. In particular, one key category we varied was culturally multilingual data, expanded across different languages in our vision dataset. As shown in Table 11, comparing results with this subset versus the full multilingual dataset highlights the impact of this specific type of data, which represents the main focus of our analysis and resource allocation.
>
> > Although the paper addresses an important problem and contributes new models to multilingual multimodal research, it offers limited architectural and methodological novelty.
>
> While our method is based on the LLaVa-Next architecture, its main contribution is the comprehensive empirical study that questions standard assumptions in constructing multilingual VLMs. Section 2.2 is not merely a description of implementation details, but the empirically derived optimal recipe resulting from our systematic ablation of training stages and components.
> Our investigation yields non-trivial methodological insights that contradict a simple "add more multilingual data" hypothesis:
> 1. Initialization Matters (Multilingual Priors vs. General Reasoning): We demonstrate that initializing with the strongest general-purpose LLMs (e.g., Gemma 2) is suboptimal for multilingual VLM performance. Despite these models possessing superior general reasoning capabilities, our experiments show that the TowerPlus backbone, which is specifically optimized for multilingualism, outperforms them. This establishes a crucial methodological finding: strong multilingual textual priors are more critical than general reasoning capabilities when constructing multilingual VLMs, a trade-off that prior work has not clearly delineated.
> 2. Stage-Specific Data Composition: We identify that simply scaling multilingual data across all stages is detrimental. Our ablation study (Section 4) reveals that English-only captions are superior for the projector alignment phase, whereas multilingual data is only beneficial during fine-tuning. This finding provides a precise "where and how" guide for multilingual data integration, refuting the idea that our approach is a trivial extension of existing pipelines.
> 3. We challenge the assumption that multilingual visual encoders are the default choice for multilingual VLMs. We found that strong, English-centric vision encoders (specifically SigLIP1) paired with high-quality multilingual data can reach the same performance of models explicitly trained with prior multilingual data.
> Therefore, our work contributes a validated methodology for constructing multilingual VLMs that challenges current "scale-up" intuitions, alongside the release of the TowerVision model and dataset.

---

> ### Author Response · Authors · 2025-11-21
> **Answer to reviewer yWBs (part2)**
>
> > The clarity of writing could be improved, as some important details are missing.
>
> Thank you for the suggestions, we answer each of your points below.
>
> > Table 4: The Gemma2-2B/9B-IT models are not included in the comparison. How do the results compare to fine-tuning the IT model?
>
> We trained two additional versions based on the Gemma2-2B/9B-IT models. Tower+ still outperforms Gemma2-IT, highlighting the complementary roles of multilingual pretraining and instruction tuning, as well as the importance of backbone selection. The updated results are included below:
>
> | Backbone Model    | TextVQA | CRBench | CC-OCR | ALM-Bench (en) | ALM-Bench (multi) |
> |------------------|---------|---------|--------|----------------|------------------|
> | Gemma2-pt-2B     | 69.2    | 61.2    | 45.3   | 74.3           | 76.7             |
> | Tower+pt-2B      | 70.3    | 62.1    | 46.3   | 73.0           | 78.2             |
> | Gemma2-it-2B     | 70.0    | 63.0    | 45.9   | 75.0           | 75.1             |
> | Tower+it-2B      | 68.1    | 58.6    | 46.1   | 77.1           | 81.1             |
> | Gemma2-pt-9B     | 72.4    | 66.6    | 49.6   | 79.9           | 79.6             |
> | Tower+pt-9B      | 73.2    | 64.5    | 54.5 | 81.3           | 84.4             |
> | Gemma2-it-9B     | **74.4** | 67.2    | 49.5   | 79.6           | 81.5             |
> | Tower+it-9B      | 73.6    | **69.7** | **56.3**   | **83.6**       | **85.2**         |
>
> > Table 5: Details about the dataset used for evaluation are missing. What data was used?
>
> We clarified that all evaluations are based on ALM-Bench. Table 5 now explicitly states the dataset used for each metric.
>
> > Line 398: The term “low-data regimes” should be clarified.
>
> We clarified that “low-data regimes” refers to scenarios where only a limited number of training data is available for fine-tuning the VLM.
>
> > Table 7: How does multilingual data affect video fine-tuning for the 9B model?
>
> Due to computational constraints, we did not evaluate the impact of multilingual data on the 9B video models.

---

> > ### Comment · Reviewer_yWBs · 2025-11-21
> > **Re authors**
> >
> > Thank you very much for additional experiments and clarifications.
> >
> > Re: "..... culturally multilingual data, expanded across different languages in our vision dataset. As shown in Table 11"
> > --> Would you please clarify what does *culturally multilingual data* mean?
> > Further, Table 11, what are the languages mean in the first column and what does the * sign mean?

---

> ### Author Response · Authors · 2025-11-25
> **reply**
>
> > Thank you very much for additional experiments and clarifications.
> Re: "..... culturally multilingual data, expanded across different languages in our vision dataset. As shown in Table 11" --> Would you please clarify what does culturally multilingual data mean? Further, Table 11, what are the languages mean in the first column and what does the * sign mean?
>
> Thank you for your helpful comments.
> Regarding the term “culturally multilingual data”: We use this term to describe cultural data originating from multiple non-English countries and cultural contexts.
> This dataset includes:
>
> - culturally significant entities and objects;
> - region-specific visual attributes;
> - biographical and historical references;
> - and other culturally grounded properties;
>
> In this case, the dataset spans multiple languages and multiple cultures, allowing us to evaluate models on culturally diverse visual understanding.
>
> Regarding Table 11 (languages and the asterisk):
>
> - The first column lists languages included in the ALM-Bench evaluation.
> - The reported values represent the average score on the language subset of the benchmark.
> - The * symbol marks languages that were not seen during training. We include them to measure the model’s ability to generalize to unseen languages when trained with expanded multilingual data.
>
>
> We have revised the text around Table 11 to make these points clearer and have updated the pdf accordingly

---

> > ### Author Response · Authors · 2025-11-28
> >
> > Dear reviewer,
> >
> > As the discussion period ends in a few days (December 3rd), we wish to follow up and confirm whether our clarifications have addressed your concerns. We would greatly appreciate any further engagement or a reconsideration of your assessment if the additional information resolves the issues you raised. Thank you

---

### Official Review · Reviewer_u13y · 2025-11-01

**Soundness:** 2
**Presentation:** 2
**Contribution:** 1
**Rating:** 2
**Confidence:** 4

**Summary:**

Most Vision-Language Models (VLMs) are English-centric and struggle to generalize across languages and cultures. This paper introduces TowerVision and TowerVideo as two variants for building and analyzing multilingual vision-language models based on the Tower+ text backbone. The approach is supported by a new multilingual dataset, VisionBlocks, that combines translated, synthetic, and culturally grounded image-text pairs. The resulting models show improved cross-lingual understanding and cultural grounding on several multilingual benchmarks. The study provides a systematic investigation of multilinguality in VLMs and explores when and how to introduce multilinguality into them.

**Strengths:**

1. The paper is well motivated by the imbalance in current VLMs, which are predominantly English-centric. Addressing multilingual and cultural inclusivity is both timely and impactful, supporting fairer and broader use of VLMs across language communities.
2. The introduction of VisionBlocks, a curated multilingual vision-language dataset, is a meaningful contribution. It provides a useful resource that directly improves multilingual performance and could benefit future research in this area.
3. Beyond presenting a multilingual model, the paper offers several valuable observations, such as the effects of multilingual backbones, alignment stages, and fine-tuning strategies, that are informative not only for multilinguality but also for the general design of VLMs.

**Weaknesses:**

1. Section 2.2 primarily describes implementation details and training configurations rather than introducing a distinct modeling contribution. It remains unclear what is newly proposed in this work versus what follows established multilingual VLM pipelines from prior studies. Thus, from a technical standpoint, the work mainly extends existing pipelines with additional multilingual data. While the dataset itself is valuable, the overall improvement, training on more data with translated text in target languages, is expected rather than conceptually new. The contribution thus lies more in resource aggregation than in methodological innovation.
2. The proposed multilingual dataset is largely an LLM-augmented collection of existing image-text and video-text resources. Several design choices, such as using the Tower model for translation, evaluating with COMETKIWI, and adopting an arbitrary threshold of 0.8 are not well justified or empirically supported. These decisions appear somewhat ad hoc and lack evidence beyond internal preference or prior use by the same research group.
3. As the authors reported, the proposed models do not outperform baselines across all tasks. Results in Table 1 and Table 3 show mixed or marginal improvements, including multilingual video-to-text tasks, suggesting that the benefits are uneven across modalities and benchmarks.
4. The analysis remains largely quantitative, focusing on benchmark scores and metrics. Given the mixed quantitative results, a qualitative examination, such as cross-lingual error cases or culturally specific examples, would provide stronger insights into the actual behavior and limitations.

**Questions:**

The main concerns stand on the technical contribution, which appears limited beyond the introduction of new multilingual data. The results are also mixed. Additional qualitative or error analyses could help explain this. While minor, please double-check citation formatting (e.g., line 148).

---

> ### Author Response · Authors · 2025-11-21
> **Answer to reviewer u13y (part1)**
>
> Thank you for your comments. Most of the raised concerns seem to be related to novelty; we answer each of your points below.
>
> > Section 2.2 primarily describes implementation details and training configurations rather than introducing a distinct modeling contribution. It remains unclear what is newly proposed in this work versus what follows established multilingual VLM pipelines from prior studies. Thus, from a technical standpoint, the work mainly extends existing pipelines with additional multilingual data. While the dataset itself is valuable, the overall improvement, training on more data with translated text in target languages, is expected rather than conceptually new. The contribution thus lies more in resource aggregation than in methodological innovation.
>
> We respectfully disagree with the assessment that our work is primarily a resource aggregation effort without methodological contributions. While we adopt the LLaVa-Next architecture as a recipe for the training pipeline, our core contribution lies in the comprehensive empirical study that challenges standard assumptions for building multilingual VLMs. Section 2.2 is not merely a description of implementation details, but the empirically derived optimal recipe resulting from our systematic ablation, revealing insights largely absent in prior work, where several design choices are assumed without extensive evaluation. Our investigation yields non-trivial methodological insights that contradict a simple "add more multilingual data" hypothesis:
> 1. Initialization Matters (Multilingual Priors): We demonstrate that initializing with the strongest general-purpose LLMs (e.g., Gemma 2) is suboptimal for multilingual VLM performance. Despite these models possessing superior general reasoning capabilities, our experiments show that the TowerPlus backbone, which is specifically optimized for multilingualism, outperforms them. This establishes a crucial methodological finding: strong multilingual textual priors are more critical than general reasoning capabilities when constructing multilingual VLMs, a trade-off that prior work has not clearly delineated.
> 2. Stage-Specific Data Composition: We identify that simply scaling multilingual data across all stages is detrimental. Our ablation study (Section 4) reveals that English-only captions are superior for the projector alignment phase, whereas multilingual data is only beneficial during fine-tuning. This finding provides a precise "where and how" guide for multilingual data integration, refuting the idea that our approach is a trivial extension of existing pipelines.
> 3. We challenge the assumption that multilingual visual encoders are the default choice for multilingual VLMs. We found that strong, English-centric vision encoders (specifically SigLIP1) paired with high-quality multilingual data can reach the same performance of models explicitly trained with prior multilingual data.
> Therefore, our work contributes a validated methodology for constructing multilingual VLMs that challenges current "scale-up" intuitions, alongside the release of the TowerVision model and dataset.

---

> > ### Author Response · Authors · 2025-11-21
> > **Answer to reviewer u13y (part2)**
> >
> > > The proposed multilingual dataset is largely an LLM-augmented collection of existing image-text and video-text resources. Several design choices, such as using the Tower model for translation, evaluating with COMETKIWI, and adopting an arbitrary threshold of 0.8 are not well justified or empirically supported. These decisions appear somewhat ad hoc and lack evidence beyond internal preference or prior use by the same research group.
> >
> > We address the concerns regarding our dataset construction and filtering choices, which are driven by specific performance metrics rather than internal preference. Regarding the translation model, we selected Tower and TowerPlus-9B over the NLLB-3.3B [1] model used in prior works (e.g., Aya-Vision, Pangea) because Tower offers objectively superior translation quality for the languages targeted in this study. This is supported by the benchmarks presented in Table 1 of Tower paper [2], where TowerInstruct outperforms the significantly larger NLLB-54B across multiple evaluation metrics.
> > Similarly, our use of COMETKiwi is grounded not only in its reference-free design but also in its empirical success in top shared tasks. For instance, in the WMT 2023 QE Shared task, Unbabel-IST COMETKiwi system [3] achieved up to +10 Spearman correlation over the previous state-of-the art. Moreover, in the WMT 2024 QE shared task, COMETKiwi continued to be a core part of strong submissions, demonstrating its robustness and strong alignment with human judgments.
> > Finally, our filtering threshold aligns with established multimodal models precedents in recent top-tier publication: Spire [4] utilized a COMETKiwi score of 0.85, and Aya-Vision, an established multilingual VLM [5],  used 0.8. Adopting these validated threshold ensures our data quality matches the standard set by these peer-reviewed works.
> >
> > [1] Costa-jussà et al., "No Language Left Behind: Scaling Human-Centered Machine Translation," 2022
> >
> > [2] Alves et al., "Tower: An Open Multilingual Large Language Model," 2024
> >
> > [3] Rei et al., "Unbabel-IST 2023 Submission for the Quality Estimation Shared Task," WMT 2023
> >
> > [4] Ambilduke et al., "From Tower to Spire: Adding the Speech Modality to a Translation-Specialist LLM" Findings of EMNLP 2025
> >
> > [5] Dash et al., "Aya Vision: Multilingual Visual Instruction Tuning”
> >
> > > As the authors reported, the proposed models do not outperform baselines across all tasks. Results in Table 1 and Table 3 show mixed or marginal improvements, including multilingual video-to-text tasks, suggesting that the benefits are uneven across modalities and benchmarks.
> > We acknowledge that performance improvements are not uniform across all benchmarks; however, this is a known trade-off when optimizing for broad multilingual generalization versus English-centric specialization. We believe the "mixed" results actually highlight the specific strengths of our approach:
> > 1. Cultural vs. General Benchmarks (Table 1): While improvements on standard English-centric benchmarks may appear marginal, Table 1 demonstrates that TowerVision significantly outperforms baselines on culturally grounded tasks (e.g., ALM-Bench) even for the smaller model. This validates our core hypothesis: standard VLMs struggle with non-Western cultural concepts, a gap our methodology explicitly bridges. The "marginal" gains in general metrics are outweighed by the substantial qualitative gains in handling diverse cultural contexts.
> > 2. Video Modality (Table 3): Regarding TowerVideo, our goal was to demonstrate the transferability of our multilingual recipe. Table 3 confirms that our text-centric multilingual alignment generalizes to the video modality with minimal adaptation. The fact that we achieve competitive results without expensive video-specific pre-training is a key finding for resource-efficient multilingual VLM development.
> >
> > > The analysis remains largely quantitative, focusing on benchmark scores and metrics. Given the results, a qualitative examination, such as cross-lingual error cases or culturally specific examples, would provide stronger insights into the actual behavior and limitations.
> >
> > This is a great suggestion. We agree that a qualitative analysis would provide deeper insights into the model’s behavior and limitations. We will incorporate a version of TowerVision that includes such cross-lingual and culturally specific examples.

---

> > > ### Comment · Reviewer_u13y · 2025-11-27
> > >
> > > Thank you for the response, the clarification of the contributions, and the additional rationale supporting the design choices. That said, I share the same concerns as the other reviewers regarding the overall novelty. Given the efforts and commitments to the new version, I have increased my score to reflect my updated understanding.

---

### Official Review · Reviewer_dxgA · 2025-11-04

**Soundness:** 3
**Presentation:** 3
**Contribution:** 2
**Rating:** 4
**Confidence:** 4

**Summary:**

The paper introduces TOWERVISION, a family of open multilingual CLMs designed to address the limitations of current English-centric VLMs . The authors also release TOWERVIDEO, a video-capable variant, and VISIONBLOCKS, a curated multilingual multimodal dataset. TOWERVISION utilizes a multi-stage training process and specific architectural choices to enhance multilingual capabilities. To overcome the scarcity of high-quality multilingual vision-text data, the authors created VISIONBLOCKS, a dataset containing approximately 6 million samples. It aggregates filtered public data, newly translated captions (using TOWER), and synthetic data generated via Gemini 2.5 to improve fine-grained visual details. TOWERVISION achieves competitive or superior performance on multilingual benchmarks like ALM-Bench (cultural understanding) and Multi30K (multimodal translation). Using a multilingual text backbone (TOWER+) consistently outperforms standard backbones (GEMMA2) for cross-modal tasks.

**Strengths:**

1. TOWERVISION achieves competitive or superior performance on various multilingual benchmarks, showing particular strength in culturally grounded tasks (like ALM-Bench) and multimodal translation (like Multi30K).

2. The paper provides a systematic study of multilingual design choices, analyzing the impact of different components such as training data composition, encoder selection, and text backbones.

3. The paper demonstrates that using a multilingual text backbone (TOWER+) and a multilingual vision encoder (SigLIP2) significantly improves cross-modal and cross-lingual performance compared to English-centric baselines .

4. The training approach substantially improves cross-lingual generalization, benefitting both high-resource and underrepresented languages, and even showing zero-shot improvements for unsupported languages.

5. The paper contributes open resources to the community, including the TOWERVISION and TOWERVIDEO models, training recipes, and VISIONBLOCKS, a curated high-quality multilingual vision-language dataset of approximately 6 million samples.

6. The work extends beyond static images to video with TOWERVIDEO, which achieves competitive performance on culturally diverse video benchmarks.

**Weaknesses:**

1. The finding that adding high-quality multilingual captions during the projector alignment stage yields "little to no positive effect" and sometimes "slightly decreases performance" on multilingual subsets is counter-intuitive to the paper's central claim that multilingual components improve performance. The paper currently relies on a surface-level hypothesis that focusing on "diverse, high-quality English captions" is simply more effective for standardizing visual and textual representations at this stage. This leaves a critical question unanswered: why does multilingual data fail at this specific stage when it succeeds elsewhere?

2. While TOWERVIDEO is presented as a major contribution, its implementation appears to be a straightforward extension of standard practices rather than a novel multilingual video approach. There is no ablation study for video-specific parameters equivalent to the depth provided for image resolution and tiling, leaving it unclear if 32 frames is optimal for multilingual video understanding, which often relies on reading temporally disparate text or recognizing subtle cultural cues.

3. The authors explicitly acknowledge that TOWERVISION is "less competitive on OCR-related tasks" due to a lack of OCR-focused data in VISIONBLOCKS. For a model aiming at robust multilingual capability, OCR is a fundamental requirement (e.g., reading menus, signs, or documents in various languages). Failing to address this known limitation weakens the claim of a truly comprehensive multilingual VLM.

4. The claim that VISIONBLOCKS is "high-quality" heavily relies on automated filtering, specifically using a COMETKIWI threshold of 0.85 for translations and Gemini 2.5 for synthetic data generation. While standard, these methods can miss subtle artifacts or cultural inaccuracies that automated metrics cannot detect, potentially undermining the "culturally grounded" goal of the model.

**Questions:**

See weaknesses

---

> ### Author Response · Authors · 2025-11-21
> **Answer to reviewer dxgA (part1)**
>
> We thank the reviewer for their time, effort and insightful comments.
>
> > The finding that adding high-quality multilingual captions during the projector alignment stage yields "little to no positive effect" and sometimes "slightly decreases performance" on multilingual subsets is counter-intuitive to the paper's central claim that multilingual components improve performances on a surface-level hypothesis that focusing on "diverse, high-quality English captions" is simply more effective for standardizing visual and textual representations at this stage. This leaves a critical question unanswered: why does multilingual data fail at this specific stage when it succeeds elsewhere?
>
> While we also found this result somewhat surprising, we think it is actually an interesting finding worth reporting. We disagree that it compromises the paper’s central claim – one of the goals of our paper is precisely to increase understanding about on which stages multilinguality leads to improvements. Our results with this experiment suggest that adding multilingual captions during projector alignment does not improve multilingual performance, and we posit that this is because the objective of this stage is fundamentally different from later fine-tuning stages.
> We interpret this as a stage-specific effect: the projector’s goal is to align visual features with the language model’s embedding space, not necessarily to expand linguistic coverage. Since the underlying LLM embeddings are typically trained on English-dominant data, using high-quality English captions provides a more homogeneous signal for alignment. By contrast, multilingual captions may introduce additional embedding variability that can disrupt this early-stage mapping (typically learned by a small MLP), even though multilingual exposure remains beneficial in later fine-tuning stages.
> In fact, this tendency to favor high-quality English captions at the alignment stage is also reported in other multilingual VLMs such as Pangea, reinforcing our motivation to conduct quantitative experiments to examine what happens when multilingual data is used instead. Our results ultimately align with these prior observations.
>
>
> > While TOWERVIDEO is presented as a major contribution, its implementation appears to be a straightforward extension of standard practices rather than a novel multilingual video approach. There is no ablation study for video-specific parameters equivalent to the depth provided for image resolution and tiling, leaving it unclear if 32 frames is optimal for multilingual video understanding, which often relies on reading temporally disparate text or recognizing subtle cultural cues.
>
> Thank you for the suggestion. We will run additional ablations for TowerVideo, in addition to those we already performed on TowerVision, such as varying image resolution, tiling, and different backbone components for the VLM.
>
>
> > The authors explicitly acknowledge that TOWERVISION is "less competitive on OCR-related tasks" due to a lack of OCR-focused data in VISIONBLOCKS. For a model aiming at robust multilingual capability, OCR is a fundamental requirement (e.g., reading menus, signs, or documents in various languages). Failing to address this known limitation weakens the claim of a truly comprehensive multilingual VLM.
>
> This is a great suggestion. We will incorporate a version of TowerVision with OCR data in the final version of the paper.

---

> > ### Author Response · Authors · 2025-11-21
> > **Answer to reviewer dxgA (part2)**
> >
> > > The claim that VISIONBLOCKS is "high-quality" heavily relies on automated filtering, specifically using a COMETKIWI threshold of 0.85 for translations and Gemini 2.5 for synthetic data generation. While standard, these methods can miss subtle artifacts or cultural inaccuracies that automated metrics cannot detect, potentially undermining the "culturally grounded" goal of the model.
> >
> > This is an important observation, and we acknowledge the limitations of relying on automated filtering. Nonetheless, we took some steps to mitigate these issues. For the translation component, we follow the strategy used by established multilingual VLMs such as Aya-Vision [1], where COMET is employed to assess translation quality during synthetic data creation. Prior work typically sets the threshold around 0.8; we increased it slightly to further reduce the likelihood of subtle translation errors slipping through. However, the premise of our work is to understand how multilingualism influences different components of the VLM pipeline, so we prioritized studying how multilingual data interacts with different training strategies over fine-grained exploration of translation-quality cutoffs.
> > For synthetic data generation, we selected Gemini 2.5 Pro after internal comparisons, as it delivers consistently strong translations while remaining scalable and cost-effective. That said, we fully recognize that even high-performing machine translation systems still struggle with nuanced or culturally grounded content. This remains an open challenge for multilingual VLM data construction more broadly.
> >
> > [4] Dash et al., "Aya Vision: Advancing the Frontier of Multilingual Multimodality"

---

> > > ### Author Response · Authors · 2025-11-27
> > > **Requested OCR Data Results**
> > >
> > > We thank the reviewer for highlighting the OCR limitation. To address this, we trained TowerVision at 9B scale using approximately 425k OCR-focused samples (about 9% of the dataset), primarily collected from FineVision, balanced between English and multilingual data.
> > >
> > > | Benchmark          | TowerVision9B | TowerVision9B + OCR |
> > > |-------------------|----------------|-----------------------|
> > > | TextVQA           | 73.6           | 76.1 ↑               |
> > > | OCRBench          | 69.7           | 72.7 ↑               |
> > > | CC-OCR            | 56.3           | 65.1 ↑               |
> > > | ALM-Bench (en)    | 83.6           | 86.1 ↑               |
> > > | ALM-Bench (multi) | 85.2           | 84.8 ↓               |
> > > | Multi30K          | 90.1           | 90.1 ↑               |
> > > | CoMMuTE           | 77.3           | 77.1 ↓               |
> > >
> > > Incorporating OCR data significantly improves OCR-centric benchmarks, however it seems to have a minor drop on some multilingual evaluation tasks (e.g., ALM-Bench multi: 85.2 → 84.8 for 9B), but overall performance remains strong.
> > > Thus, TowerVision with OCR data achieves substantial OCR improvements while maintaining robust multilingual capabilities.

---

> > > > ### Author Response · Authors · 2025-11-28
> > > >
> > > > Dear reviewer,
> > > >
> > > > As the discussion period ends in a few days, we wish to follow up and confirm whether our clarifications have addressed your concerns. We would greatly appreciate any further engagement or a reconsideration of your assessment if the additional information resolves the issues you raised. Thank you

---

### Author Response · Authors · 2025-11-21
**Word of Appreciation**

We sincerely thank you for taking the time to read and provide detailed feedback on our submission.

Your thoughtful reviews are greatly appreciated, and your insights are helping us improve our work.

We have carefully addressed each claim raised and hope that our responses clarify any questions or concerns.

---

### Author Response · Authors · 2025-12-03
**Reviewer Feedback Summary for the AC**

Reviewer **dxgA**  noted that (1) multilingual captions slightly hurt performance during projector alignment, (2) that TOWERVIDEO lacked ablations, (3) OCR performance was limited, and (4) VISIONBLOCKS relied heavily on automated filtering. Our answer clarifies  that the projector alignment effect is stage-specific: English captions provide a cleaner signal for embedding alignment, while multilinguality benefits later fine-tuning. We also  incorporated ~425k OCR-focused samples into TowerVision-9B achieving **strong OCR improvements**, and  we provided details to back up our choice of stricter thresholds (0.85), which aligns with prior work and reinforces our emphasis on high-quality data for understanding multilingual interactions.

Reviewer **u13y** questioned the novelty of our work and our choices for dataset construction . We clarified that our core contribution  is a systematic empirical study revealing non-trivial insights: stage-specific data composition (English captions for alignment, multilingual later), TowerPlus backbone outperforming larger general-purpose LLMs for multilingual tasks, and English-centric vision encoders matching multilingual encoders when combined with high-quality multilingual data. (We justified dataset choices (Tower/TowerPlus, COMETKiwi, threshold 0.85) with empirical evidence and prior work, highlighted strong gains on culturally grounded tasks and video transfer, and committed to adding cross-lingual and culturally specific qualitative examples. **As can be seen in the comments, this reviewer acknowledged our response and raised the score.**

Reviewer **yWBs** suggested a (1) deeper analysis of data mixtures, (2) noted limited architectural and methodological novelty, requested (3) clarity on missing details in tables, and asked (4) for clarification on culturally multilingual data. We noted in (1) that our analysis of data mixtures primarily focused on culturally grounded multilingual data, which was the central focus of the study, (2) emphasized methodological insights beyond “more data” (stage-specific data composition, TowerPlus backbone, English-centric vision encoders), (3) added comparisons with Gemma2-IT models, (4) clarified datasets and evaluation details, explained “low-data regimes,” and defined culturally multilingual data as content from multiple non-English countries/cultures.


**Final Summary for AC**

We believe our revisions fully addressed all conceptual and methodological questions raised by the reviewers, expanded our experimental analysis, and clarified details about  the dataset. We also report new experiments increasing the OCR traning data, leading to substantial improvements in OCR tasks, as suggested by one of the reviewers. All the reviewers acknowledged these improvements in their discussion and one of them (R2) increased their score to reflect this updated understanding.
Overall, our revisions resolved all substantive points raised by the reviewers.
We believe the paper meets the bar for ICLR acceptance, offering (1) a practical and reproducible multilingual benchmark, (2) novel empirical insights into alignment-based multilinguality, and (3) strong ablations and cross-model validations useful for the multilingual VLM community. **We stress that our proposed model, TowerVision, achieves performance on par with or surpassing state-of-the-art models like Qwen2.5-VL, Gemma3, and Aya-Vision, which are not fully open, and sets a new standard on translation, multicultural, and culturally grounded tasks** as demonstrated on benchmarks like ALM-Bench and Multi30k, highlighting its effectiveness across diverse multilingual contexts.

Thank you.

---

### Author Response · Authors · 2025-12-03

To facilitate the AC’s assessment, we provide a concise summary of the main concerns of reviewers and our responses, highlighting how the reviewers’ concerns were addressed. All reviewers found the paper clear and well-motivated, emphasizing the importance of addressing the imbalance in current VLMs, which remain predominantly English-centric. They agreed that our work provides (1) a systematic study of multilingual design choices and that (2) improving multilingual and cultural inclusivity is both timely and impactful, supporting fairer and broader deployment of VLMs across diverse language communities.

**Main Concern: Architectural Novelty**

A common concern was that the overall architecture largely follows existing VLM pipelines and may appear lacking in novelty. While our design intentionally builds on LLaVA-Next, we clarified to the reviewers that the primary contribution of our work is not architectural innovation, but a comprehensive empirical investigation that systematically questions standard assumptions in multilingual VLM construction. Section 2.2 is not implementation detail, it reflects an empirically derived recipe obtained through controlled ablations of data allocation, pretraining stages, and initialization strategies. Our findings surface several non-trivial insights:
1. Initialization Matters: Multilingual priors outperform general reasoning. Models initialized from multilingual LLMs (e.g., TowerPlus) consistently beat strong general-purpose LLMs like Gemma-2.
2. Stage-Specific Data Composition: Multilingual data is not uniformly helpful.
   . English-only captions are optimal for projector alignment.
   . Multilingual data provides gains primarily during instruction tuning.
3. Vision Encoder Choice: English-centric encoders (SigLIP-1) combined with high-quality multilingual data match or surpass explicitly multilingual vision encoders, countering prior assumptions.


Together, these results provide a validated methodology for multilingual VLM construction, challenging common scale-centric intuitions.
We also release the TowerVision model and dataset to support future research, and the manuscript has been revised to make this perspective explicit.

---

### Note · Authors · 2026-03-27

I have read and agree with the venue's withdrawal policy on behalf of myself and my co-authors.

---

### Meta-Review · Area_Chair_Zobj · 2025-12-12

**Summary:**

Unanimity of the reviewers on the negative side. The authors provided responses to a number of concerns.
The ACs carefully read the paper, the reviews, and the authors' responses.
The paper requires major revision and a second review round.

**Reviewer Concerns:**

Unanimity of the reviewers on the negative side. The authors provided responses to a number of concerns.
The ACs carefully read the paper, the reviews, and the authors' responses.
The paper requires major revision and a second review round.

**Reviewer Scores:**

Unanimity of the reviewers on the negative side. The authors provided responses to a number of concerns.
The ACs carefully read the paper, the reviews, and the authors' responses.
The paper requires major revision and a second review round.

---

### Decision · Program_Chairs · 2026-01-26

Reject